# INDIGO: GNN-Based Inductive Knowledge Graph Completion Using Pair-Wise Encoding

**Shuwen Liu[1], Bernardo Cuenca Grau[1], Ian Horrocks[1], and Egor V. Kostylev[2]**

[1]Department of Computer Science, University of Oxford, UK

{shuwen.liu, bernardo.cuenca.grau, ian.horrocks}@cs.ox.ac.uk

[2]Department of Informatics, University of Oslo

egork@ifi.uio.no

## Abstract

The aim of knowledge graph (KG) completion is to extend an incomplete KG with missing triples. Popular approaches based on graph embeddings typically work by first representing the KG in a vector space, and then applying a predefined scoring function to the resulting vectors to complete the KG. These approaches work well in *transductive* settings, where predicted triples involve only constants seen during training; however, they are not applicable in *inductive* settings, where the KG on which the model was trained is extended with new constants or merged with other KGs. The use of Graph Neural Networks (GNNs) has recently been proposed as a way to overcome these limitations; however, existing approaches do not fully exploit the capabilities of GNNs and still rely on heuristics and ad-hoc scoring functions. In this paper, we propose a novel approach, where the KG is fully encoded into a GNN in a transparent way, and where the predicted triples can be read out directly from the last layer of the GNN without the need for additional components or scoring functions. Our experiments show that our model outperforms state-of-the-art approaches on inductive KG completion benchmarks.

## 1 Introduction

Knowledge graphs (KGs) are graph-structured knowledge bases where nodes and edges represent entities of interest and their relations [7]. KGs are commonly represented as sets of *triples* in a standard format such as the Resource Description Framework (RDF) [9]. Many prominent KGs are highly incomplete, which limits their usefulness in practice; hence KG *completion* (or *link prediction*)—the problem of extending a KG with missing triples—has received significant attention [14].

Most approaches to KG completion, such as TransE [3], DistMult [28], and RotatE [20], are based on graph embedding techniques, which first embed the KG into a vector space (e.g., by learning a feature vector for each entity) and then generate the predicted triples by applying a predefined scoring function to the learnt vectors [14]. These models demonstrate good performance in *transductive* settings, where missing triples are assumed to mention only constants already occurring in the incomplete KG. A key limitation of these models, however, is that they are not applicable in *inductive* settings [5, 24, 21], where missing triples may involve constants unseen during training; this setting is especially relevant in practice since KGs are evolving: they may be extended with triples describing new objects or integrated with external KGs. Consider, for instance, the KG inside the frame in Fig. 1, where an embedding-based model has been used to extend the KG with the triple (*Plato*, lives, *Greece*); thus, the model seems to have successfully learnt the pattern that students and their teachers tend to live in the same country. Assume the graph is now extended with the triples

$$(\textit{Aristotle}, \text{student}, \textit{Plato}), \quad (\textit{R.Feynman}, \text{student}, \textit{J.Wheeler}), \quad (\textit{J.Wheeler}, \text{lives}, \textit{USA}),$$

35th Conference on Neural Information Processing Systems (NeurIPS 2021).

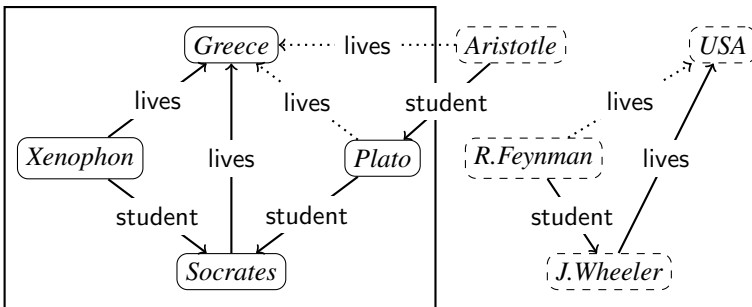

Figure 1: An example for link prediction, where solid and dotted arrows represent known and missing triples, respectively. If we use the graph enclosed within the frame for training, transductive methods will only be able to predict the missing triple (*Plato*, lives, *Greece*) inside the frame; in contrast, inductive methods can take the new information (e.g., triple (*Aristotle*, student, *Plato*)) into account and predict all three missing triples without re-training.

introducing constants for Aristotle, Feynman, Wheeler, and the USA. Since the parameters learnt by embedding-based models are tied to the constants in the original KG, such a model cannot generalise the learnt pattern to the constants unseen during training, and fails to complete the KG with triples (*Aristotle*, lives, *Greece*) and (*R.Feynman*, lives, *USA*). Indeed, for the model to become applicable, it would require re-training using also the newly introduced triples.

There are various approaches for inductive KG completion. Rule-based methods [29, 10, 15] show good results, but only when the shape of the rules is known in advance. Other methods utilise side information about unseen constants, such as attributes [6] and textual descriptions [25, 18], which is not always available in practice. Recent approaches [5, 17, 30, 24, 21] aim to overcome these limitations by exploiting *Graph Neural Networks* (*GNNs*), which are directly applicable to graphs annotated with feature vectors [16]. GNNs are biased towards graph symmetries (nodes with same neighbourhoods receive the same value) and have built-in inductive capabilities: they are invariant under isomorphisms (i.e., insensitive to the identity of nodes) and can capture general patterns represented as logic formulas [2, 11, 27].

Conceptually, we see GNN-based KG completion approaches as working in three stages. First, they *encode* the input KG as a graph with nodes annotated by feature vectors; second, this graph is fed to a GNN; third, the predicted triples are *decoded* from the output vectors of the GNN. Approaches differ in their encoding of the KG, the kind of GNN they use, and the chosen function for decoding. In the R-GCN approach [17], the encoder generates a node with a randomly initialised vector for each entity in the KG and triples are encoded as directed coloured edges (colours correspond to binary relations); their GNN model aggregates separately for each colour, and a colour-specific scoring function is applied to the outermost GNN layer for decoding the prediction. Hamaguchi et al. [5] and Wang et al. [24] developed similar approaches with decoding also based on a scoring function. A drawback of these approaches is that their prediction for a triple depends on the standalone neighbourhoods of its constants in the input KG, but does not take into account what is the common part of these neighbourhoods. This drawback has been recently addressed in the GraIL system by Teru et al. [21]. To predict whether a given triple should be added to the KG, GraIL identifies a sub-graph of the KG for that triple and encodes it in a way similar to R-GCN. Then, GraIL applies a GNN and makes a prediction for this triple using a scoring function applied globally to the output vectors of all nodes in its dedicated subgraph. To make predictions for many triples, the approach requires generating a subgraph to each of them, which is a significant bottleneck in practice.

In this paper, we propose a novel approach where the KG is encoded into the input graph to a graph convolutional network [8]) in a transparent and direct way, so that the inductive capabilities of the GNN are fully exploited. In contrast to existing approaches, our encoding establishes a one-to-one correspondence between elements of the feature vectors in the innermost and outermost layers of the GNN and triples over the KG's signature, and hence the predicted triples can be read out directly from the outermost layer without the need for an external scoring function. As a result, our approach has neither the aforementioned drawbacks of scoring-based approaches, nor the bottleneck of GraIL, being able to process the entire graph at once; moreover, our approach allows to make predictions for

several triples in one run, which may provide significant speed up in practice. We have implemented our approach in a system called *INDIGO* and compared it with R-GCN, GraIL and the system in [5] on the inductive benchmarks developed in [21] and [5], as well as on a new benchmark we have developed based on Freebase. Our results show that INDIGO not only outperforms the baselines on these benchmarks, but can also be trained and applied more efficiently. We have finally studied the ability of our system to learn in practice common inference patterns represented using logical rules; our experiments show that our system is able to learn more comprehensive rule sets than the evaluated baselines.

## 2 Inductive KG Completion

In the context of this paper, a knowledge graph is a set of triples represented in a standard format for graph-structured data such as RDF [9]. More formally, a *signature* consists of pairwise disjoint sets of *types* (i.e., unary predicates), *relations* (i.e., binary predicates), and *constants*. A *knowledge graph (KG)* is a finite set of triples of the form $(c, \text{type}, t)$, where $c$ is a constant and $t$ is a type, and triples of the form $(c, r, d)$, where $c$, $d$ are constants and $r$ is a relation.[1] The *signature* $\text{Sig}(\mathcal{K})$ of a KG $\mathcal{K}$ is the set of types, relations, and constants used in $\mathcal{K}$; then, $\text{Pred}(\mathcal{K})$ is the subset of $\text{Sig}(\mathcal{K})$ consisting of all its types and relations.

The problem of *KG completion* can be loosely understood as that of extending an incomplete KG $\mathcal{K}$ to its complete version $\mathcal{K}^*$ by adding triples over $\text{Sig}(\mathcal{K})$, where a triple is added if there is sufficient evidence that it holds given the triples in $\mathcal{K}$. We formalise *inductive* KG completion as the following ML problem. Given arbitrary (but fixed) finite sets Types and Rels of types and relations, respectively, the aim in *inductive* KG completion is to learn a Boolean *completion function* $f(\cdot, \cdot)$ applicable to each pair of a KG $\mathcal{K}$ with $\text{Pred}(\mathcal{K}) \subseteq \text{Types} \cup \text{Rels}$ and a triple $\lambda$ with $\text{Sig}(\lambda) \subseteq \text{Sig}(\mathcal{K})$ such that $f(\mathcal{K}, \lambda)$ is true if $\lambda$ is in $\mathcal{K}^*$. Note that, in this formalisation, *transductive* KG completion is the particular case of inductive completion where the function to be learnt is applicable only to a fixed KG $\mathcal{K}$—that is, can be seen as $f_{\mathcal{K}}(\cdot)$ such that $f_{\mathcal{K}}(\lambda)$ is true for a triple $\lambda$ if $\lambda$ belongs to $\mathcal{K}^*$.

A key limitation of transductive approaches based on graph embeddings is that the completion function learnt for a given KG is not applicable to any other KG (for this one needs to independently learn a new function). In other words, a trained instance of such an approach cannot make predictions for triples involving constants unseen during training; for example, a model trained by such a system on a graph in the frame in Fig. 1 cannot make predictions for any triple with a constant outside this frame. In contrast, once trained on a KG $\mathcal{K}$, inductive GNN-based systems, such as GraIL and our system, can make predictions without re-training on every KG and triple over the same types and relations as $\mathcal{K}$, regardless of the constants they use.

## 3 A GNN-Based Architecture for Inductive KG Completion

### 3.1 Overview

Our inductive approach relies on the completion function $f$ realised by the following three steps.

1. *Encoding*, which takes an (incomplete) KG $\mathcal{K}$ and a set $\Lambda$ of candidate triples (of the same signature) as input and returns a node-annotated graph $G_{\mathcal{K}}^{\Lambda}$ of the form specified in Definition 1; the encoding is *pair-wise*: nodes in $G_{\mathcal{K}}^{\Lambda}$ correspond to pairs of constants occurring in $\mathcal{K}$ and $\Lambda$.
2. *GNN* application, which updates the annotations of the graph nodes; in our approach, we use a *graph convolutional network (GCN)*, which is a GNN variant popular in applications [8].
3. *Decoding*, which extracts the predictions $f(\mathcal{K}, \lambda)$ for each $\lambda \in \Lambda$ from the updated graph output by the GNN; this step is essentially mirroring the encoding.

The details of these steps are given in Sections 3.2, 3.3, and 3.4, respectively. The key conceptual difference distinguishing our approach from existing inductive systems, such as R-GCN and GraIL, lies in the encoding and decoding steps. On the one hand, existing approaches typically encode each constant in $\mathcal{K}$ by a unique node in the graph, where the node's feature vector is randomly initialised; in contrast, in our approach each node in the graph encodes a *pair* of constants, and each element in

---

[1]In the KG literature, constants and types are sometimes collectively referred to as *entities* and the type relation is treated as an ordinary relation.

the initial feature vector of this node directly captures a fact in the KG involving these constants. On the other hand, existing approaches typically rely on ad-hoc scoring functions to decode the output of the GNN, whereas in our approach the predicted triples can be read out *directly* from the feature vectors in the final layer of the GNN. Furthermore, in our approach, we are able to make predictions for a set $\Lambda$ of candidate triples at once (for the same KG) rather than for a single triple; this facilitates training and testing in many practical scenarios, including those reflected in existing benchmarks.

Next, we define the node-annotated graphs which are used on all three steps of our approach. There are significant differences between KGs, which represent graph-structured data, and annotated graphs, which are the graphs manipulated by GNNs. Nodes in a KG represent constants and types, and edges are labelled with relations between such entities; in contrast, nodes in annotated graphs are annotated with feature vectors, and (undirected and unlabelled) edges between nodes indicate that the values of their vectors influence each other during the execution of the GNN.

**Definition 1.** *For annotation dimension $\delta \in \mathbb{N}$, a $\delta$-annotated graph is an undirected graph where each node $u$ is associated with a* feature vector $\mathbf{u} \in \mathbb{R}^\delta$. *The $i$-th element of $\mathbf{u}$ is denoted by $(\mathbf{u})_i$.*

In our approach, the annotation dimension $\delta$ is determined by the relevant sets Rels of relations and Types of types (which are assumed known and fixed before training in the inductive setting); in particular, we take $\delta = |\text{Types}| + 2 \cdot |\text{Rels}|$. This allows us to associate positions in feature vectors to types, relations, and inverses of relations, which will be instrumental for encoding and decoding. In particular, we fix an (arbitrary) *enumeration* id assigning a unique natural number from 1 to $\delta$ to each $t \in$ Types, each $r \in$ Rels, and the *inverse* $r^-$ of each $r \in$ Rels, and assume that, for every type, relation, or inverse $p$, element $\text{id}(p)$ of a node feature vector corresponds to $p$. Observe that the dimension $\delta$ does not depend on the constants of the input KG; so our approach is indeed inductive: once trained, our model is applicable to any KG over the fixed sets of types and relations, but using arbitrary constants. For the rest of this section, we assume that Rels, Types, id, and $\delta$ are fixed.

## 3.2 Encoding KGs

Our encoder maps a KG $\mathcal{K}$ and a set $\Lambda$ of candidate triples to a $\delta$-annotated graph $G_\mathcal{K}^\Lambda$ where each node $u_{c,d}$ corresponds to a pair of (not necessarily distinct) constants $c, d$ connected by a triple in $\mathcal{K}$ or $\Lambda$. To avoid duplication of information, graph $G_\mathcal{K}^\Lambda$ contains only one of the nodes $u_{c,d}$ and $u_{d,c}$ when $c \neq d$. The choice between $u_{c,d}$ and $u_{d,c}$ is immaterial to our approach; for definiteness, we employ the lexicographic order $\preceq$ on the set of constants and add node $u_{c,d}$ to $G_\mathcal{K}^\Lambda$ only if $c \preceq d$. As formalised in the following definition, the annotation of $u_{c,d}$ is a Boolean feature vector where an element is set to 1 if and only if the corresponding type, relation, or inverse relation holds in $\mathcal{K}$ for $c$ and $d$ (e.g., $(\mathbf{u}_{c,d})_i = 1$ if and only if $c = d$ and $(c, \text{type}, t) \in \mathcal{K}$ for type $t$ with $\text{id}(t) = i$); if $c$ and $d$ appear in the same triple only in $\Lambda$, then the elements of $\mathbf{u}_{c,d}$ are all 0. As a result, our encoding establishes a one-to-one correspondence between elements of the feature vectors and relevant triples over the KG's signature. Finally, two nodes in $G_\mathcal{K}^\Lambda$ are connected by an edge if they share a constant, which, as we will see, allows GNNs to learn structural patterns in the data (e.g., such as the pattern in Fig. 1 capturing that teachers live in the same country as their students).

**Definition 2.** *The* encoding *of a KG $\mathcal{K}$ over* Types *and* Rels*, and a set $\Lambda$ of candidate triples of the same signature as $\mathcal{K}$ is the $\delta$-annotated graph $G_\mathcal{K}^\Lambda$ where*

- *$G_\mathcal{K}^\Lambda$ has a node $u_{c,d}$ for every two constants $c, d$ in $\mathcal{K}$ such that*
  - *either $c = d$,*
  - *or $c \prec d$ and $\mathcal{K} \cup \Lambda$ contains a triple of the form $(c, r, d)$ or $(d, r, c)$ for some relation $r$;*
- *the feature vectors of nodes in $G_\mathcal{K}^\Lambda$ are defined as*
  - *$(\mathbf{u}_{c,c})_{\text{id}(t)} = 1$ for all $(c, \text{type}, t) \in \mathcal{K}$,*
  - *$(\mathbf{u}_{c,d})_{\text{id}(r)} = 1$ for all $(c, r, d) \in \mathcal{K}$ with $c \preceq d$ and*
  - *$(\mathbf{u}_{c,d})_{\text{id}(r^-)} = 1$ for all $(d, r, c) \in \mathcal{K}$ with $c \preceq d$,*
  - *all other elements are 0;*
- *$G_\mathcal{K}^\Lambda$ has an edge between different nodes $u_X$ and $u_Y$ if pairs $X$ and $Y$ have a constant in common.*

The graph in Fig. 2(a) depicts the structure of $G_\mathcal{K}^\Lambda$ for our running example KG $\mathcal{K}$ inside the frame in Fig. 1 and $\Lambda = \{(\textit{Plato}, \text{lives}, \textit{Greece})\}$ (where we assume that enumeration id of relations is lexicographic). Nodes represent pairs of constants, and two nodes are connected only if their pairs share a constant. Since our example $\mathcal{K}$ has no types and two relations, each feature vector (not

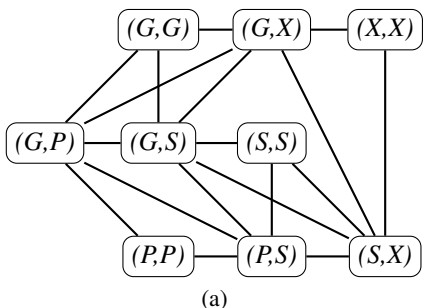 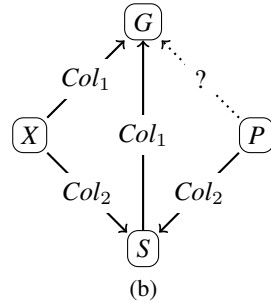

(a)              (b)

Figure 2: Encodings of the KG in the frame in Fig. 1 with a candidate triple (*Plato*, lives, *Greece*). (a) Encoding of INDIGO, where the feature vectors of the nodes are omitted and the node names are abbreviated; for example $(G, X)$ denotes $u_{Greece,Xenophon}$. (b) Encodings of R-GCN, where the names of constants are abbreviated to their first letter, and $Col_1$ and $Col_2$ are colours.

shown in the figure for clarity) has dimension $4$. For instance, the vector of $(G, X)$ is $[0, 0, 1, 0]$; the first two components are set to $0$ because $\mathcal{K}$ does not contain the triple (*Greece*, lives, *Xenophon*) or (*Greece*, student, *Xenophon*), the third is $1$ since $\mathcal{K}$ contains (*Xenophon*, lives, *Greece*), and the last is $0$ because there is no triple (*Xenophon*, student, *Greece*) in $\mathcal{K}$. As we will see in Section 3.4, our model predicts the triple (*Plato*, lives, *Greece*) to be in the completion $\mathcal{K}^*$ by examining the feature vector of $(G, P)$ after processing $G_{\mathcal{K}}^{\Lambda}$ using a GNN as in Section 3.3; in the encoding, the node $(G, P)$ is justified in $G_{\mathcal{K}}^{\Lambda}$ by $\Lambda$ and initialised with vector $[0, 0, 0, 0]$, since *Plato* and *Greece* are not connected in $\mathcal{K}$.

It is interesting to compare our encoding to that of existing approaches. In particular, Fig. 2(b) depicts the encoding of R-GCN [17] for the same example. The annotated graph now contains one node per constant, and its directed edges are labelled with colours $Col_1$ and $Col_2$, representing relations lives and student; thus, the structure of the annotated graph closely mimics that of the KG. In contrast to our approach, however, the feature vectors associated to each node do not bear any connection with the contents of the KG and are typically initialised randomly. After processing this graph by an appropriate GNN, the R-GCN model applies a scoring function to the feature vectors of *Plato* and *Greece* in the output layer of the GNN to predict the presence of the triple in the completion $\mathcal{K}^*$.

Observe that, differently to R-GCN, the size of the encoding graph in our approach can be larger than that of the original KG. Although the size growth can be quadratic in theory, we next argue that it is almost always linear in practice. To this end, let $\mathcal{K}$ be a KG and let $\Lambda$ be a set of candidate triples of the same signature. By construction, encoding $G_{\mathcal{K}}^{\Lambda}$ has $m + |\mathcal{K}| + |\Lambda|$ nodes annotated by $\delta$-vectors and up to $(m + |\mathcal{K}| + |\Lambda|) * n$ edges, where $m$ is the number of constants in $\mathcal{K}$ and $n$ is the maximal degree of a constant in $\mathcal{K} \cup \Lambda$ (here, the *degree* of a constant $c$ is the number of constants $d$ such that $c$ and $d$ appear in the same triple in $\mathcal{K} \cup \Lambda$). In theory, $n$ is bounded by $m$, which is, however, approached only if every two constants are connected by a triple in $\mathcal{K}$ or $\Lambda$. This hardly happens in practice, since real-world KGs are typically sparse. Hence, we can assume that $n$ is small and the encoding size is linear (assuming that $\delta$, which is the number of types and relations, is also small).

### 3.3 The GNN Model

Broadly speaking, a GNN is characterised by *aggregation* and *combination* functions for each layer, where the functions usually depend on learnable parameters. Every layer of the GNN updates the feature vector of each node in the graph by first aggregating the previous vectors of the neighbouring nodes using the aggregation function, and then combining the result with the node's previous vector using the combination function. The vectors in the final layer form the GNN's output.

In our approach, we use a *graph convolutional network* (*GCN*) variant [8] with ReLU activation on the hidden layers, which offers a good balance between expressivity and performance (note, however, that the choice of GCNs is not crucial for our approach, and GCNs can be easily replaced with any other suitable GNN variant). The following definition of GCNs relies on a representation of each $\delta$-annotated graph $G$ with $N$ nodes as two matrices $\mathbf{A}_G \in \{0, 1\}^{N \times N}$ and $\mathbf{U}_G \in \mathbb{R}^{N \times \delta}$, with

$\mathbf{A}_G = \mathbf{A}'_G + \mathbf{I}$ for $\mathbf{A}'_G$ the adjacency matrix of $G$ and $\mathbf{I}$ the identity matrix, and $\mathbf{U}_G$ the *feature matrix* of $G$ where each row is the feature vector of a node in $G$. This representation relies on an ordering of nodes in $G$, which we assume arbitrarily fixed for each graph. The diagonal identity matrix $\mathbf{I}$ plays a technical role: in essence, it adds an artificial loop to each node in the graph and thus allows for a uniform treatment of combination and aggregation in GCNs. In fact, instead of $\mathbf{A}_G$, GCNs use its *normalised* version $\mathbf{A}_G^{\mathsf{norm}} = \mathbf{D}_G^{-\frac{1}{2}} \mathbf{A}_G \mathbf{D}_G^{-\frac{1}{2}}$, where $\mathbf{D}_G$ is the *node degree matrix* of $\mathbf{A}_G$—that is, the diagonal matrix where the $i$-th diagonal element is defined as $(\mathbf{D}_G)_{ii} = \sum_j (\mathbf{A}_G)_{ij}$ (where $\mathbf{M}_{ij}$ is the element of a matrix $\mathbf{M}$ in row $i$ and column $j$).

**Definition 3.** *A* graph convolutional network *(GCN)* $\mathfrak{N}$ with $L \in \mathbb{N}$ layers *and* dimension $\delta \in \mathbb{N}$ *is characterised by (learnable)* weight *matrices* $\mathbf{W}_\ell \in \mathbb{R}^{\delta_{\ell-1} \times \delta_\ell}$ *and* bias *vectors* $\mathbf{b}_\ell \in \mathbb{R}^{\delta_\ell}$ *for all layers* $\ell \in \{1, \ldots, L\}$, *where each* $\delta_\ell \in \mathbb{N}$ *is the* dimension *of layer* $\ell$ *such that* $\delta_0 = \delta_L = \delta$ *(the number $L$ of layers and the dimensions $\delta_\ell$ of the hidden layers are hyper-parameters of the model). For an input $\delta$-annotated graph $G$ with $N$ nodes given as matrices $\mathbf{A}_G$ and $\mathbf{U}_G$, $\mathfrak{N}$ updates, on each layer* $\ell \in \{1, \ldots, L\}$, *its* hidden state *matrix* $\mathbf{U}_\ell \in \mathbb{R}^{N \times \delta_\ell}$ *using the rule*

$$\mathbf{U}_\ell = \sigma_\ell(\ \mathbf{A}_G^{\mathsf{norm}}\ \mathbf{U}_{\ell-1}\ \mathbf{W}_\ell + \mathbf{b}_\ell),$$

*where* $\mathbf{U}_0 = \mathbf{U}_G$ *and the non-linear* activation *function* $\sigma_\ell$ *is ReLU for each* $\ell < L$ *and the logistic sigmoid function for* $\ell = L$, *which (in both cases) is applied to matrices element-wise. Then,* $\mathfrak{N}(G)$ *is the $\delta$-annotated graph with the same nodes and edges as $G$, where the feature vector of each node $u$ is the row in* $\mathbf{U}_L$ *corresponding to* $u$.

Observe that the all the elements of a GCN, including all trainable and non-trainable parameters, do not depend on the number of nodes in the graph, and so each (trained) GCN is applicable to graphs of arbitrary size. Also, the activations are justified by practice: ReLU on hidden layers is computationally efficient and prevents the vanishing gradient problem, while sigmoid on the final layer guarantees values to be in $(0, 1)$, which is convenient for computing the loss.

## 3.4 Decoding the GNN Output

The result $\mathfrak{N}(G_\mathcal{K}^\Lambda)$ of a GCN $\mathfrak{N}$ applied to the encoding $G_\mathcal{K}^\Lambda$ of $\mathcal{K}$ and a candidate set $\Lambda$ can be decoded into a set of triples over the same signature by essentially inverting the encoder. Note, however, that features in $\mathfrak{N}(G_\mathcal{K}^\Lambda)$ are not Boolean; so, to decide if the decoded graph contains a triple, we check if the corresponding feature in $\mathfrak{N}(G_\mathcal{K}^\Lambda)$ is above a threshold $\theta$, which we take as 0.5 in our experiments.

**Definition 4.** *Given a threshold value* $\theta \in \mathbb{R}$, *the* decoding *of $\delta$-annotated graph $G$ is the set* $\mathcal{K}_{\mathsf{dec}}(G)$ *of the following triples, for $c$, $d$ constants, $t$ a type, and $r$ a relation:*

- $(c, \mathsf{type}, t)$ *such that* $(\mathbf{u}_{c,c})_{\mathsf{id}(t)} \geq \theta$,
- $(c, r, d)$ *such that* $(\mathbf{u}_{c,d})_{\mathsf{id}(r)} \geq \theta$ *and* $c \preceq d$,
- $(d, r, c)$ *such that* $(\mathbf{u}_{c,d})_{\mathsf{id}(r^-)} \geq \theta$ *and* $c \preceq d$.

Overall, a triple $\lambda$ in a candidate set $\Lambda$ for a KG $\mathcal{K}$ is predicted by a GCN $\mathfrak{N}$ to be in the completion $\mathcal{K}^*$ if $\lambda \in \mathcal{K}_{\mathsf{dec}}(\mathfrak{N}(G_\mathcal{K}^\Lambda))$. Note, however, that in training and evaluation we sometimes need not just a Boolean prediction for a candidate triple, but a *confidence* in this prediction—that is, a numeric value from $(0, 1)$; for this, we directly take the corresponding component of $\mathbf{u}_{c,d}$ in the final layer.

## 3.5 Capturing Logical Rules

In addition to using standard benchmarks, we will also compare the inductive capabilities of completion approaches in terms of their ability to capture common inference patterns represented in *Datalog* [1]—a well-known logic-based rule language in knowledge representation and databases.

A *(Datalog) rule* is a function-free first-order logic sentence of the form

$$\forall \mathbf{x}.\ B_1, \ldots, B_n \to H, \tag{1}$$

where all $B_i$ and $H$ are *atoms* over predicates in $\mathsf{Types} \cup \mathsf{Rels}$ using variables in $\mathbf{x}$ as terms, such that each variable in $H$ is mentioned in one of $B_i$; in our context, it is convenient to see atoms as triples as in a KG where variables are used instead of constants. Each assignment $\sigma$ of constants to variables $\mathbf{x}$ of a rule $r$ of form (1) maps the atoms $B_1, \ldots, B_n$ into a KG $\mathcal{K}_r^\sigma$; in the same way, $\sigma$ maps $H$ to a

Table 1: Benchmarks statistics, where $|\mathcal{T}|$, $|\mathcal{V}|$, $|\mathcal{K}_{\text{test}}|$, and $|\Lambda^+_{\text{test}}|$ are the sizes of the corresponding training set, validation set, incomplete test KG, and positive test triple set, respectively.

| | GraIL-BM / FB15K-237 | | | | GraIL-BM / NELL-995 | | | | GraIL-BM / WN18RR | | | |
|---|---|---|---|---|---|---|---|---|---|---|---|---|
| | v1 | v2 | v3 | v4 | v1 | v2 | v3 | v4 | v1 | v2 | v3 | v4 |
| $|\mathcal{T}|$ | 4,245 | 9,739 | 17,986 | 27,203 | 4,687 | 8,219 | 16,393 | 7,546 | 5,410 | 15,262 | 25,901 | 7,940 |
| $|\mathcal{V}|$ | 489 | 1,166 | 2,194 | 3,352 | 414 | 922 | 1,851 | 876 | 630 | 1,838 | 3,097 | 934 |
| $|\mathcal{K}_{\text{test}}|$ | 1,993 | 4,145 | 7,406 | 11,714 | 833 | 4,586 | 8,048 | 7,073 | 1,618 | 4,011 | 6,327 | 12,334 |
| $|\Lambda^+_{\text{test}}|$ | 205 | 478 | 865 | 1,424 | 100 | 476 | 809 | 731 | 188 | 441 | 605 | 1,429 |

| | Hamaguchi-BM | | | | | | | | | INDIGO-BM |
|---|---|---|---|---|---|---|---|---|---|---|
| | h-1K | h-3K | h-5K | t-1K | t-3K | t-5K | b-1K | b-3K | b-5K | |
| $|\mathcal{T}|$ | 108,197 | 99,963 | 92,309 | 96,968 | 78,763 | 67,774 | 93,364 | 71,097 | 57,601 | 121,601 |
| $|\mathcal{V}|$ | 4,613 | 4,184 | 3,845 | 3,999 | 3,122 | 2,601 | 3,799 | 2,759 | 2,166 | 14,121 |
| $|\mathcal{K}_{\text{test}}|$ | 4,352 | 12,376 | 19,625 | 15,277 | 31,770 | 40,584 | 18,638 | 38,285 | 48,425 | 250,195 |
| $|\Lambda^+_{\text{test}}|$ | 994 | 2,969 | 4,919 | 986 | 2,880 | 4,603 | 960 | 2,708 | 4,196 | 14,904 |

triple $t^\sigma_r$. A completion function (or a GNN-based model realising this function) *captures* $r$ if, for every assignment $\sigma$ of constants to $\mathbf{x}$, the model predicts triple $t^\sigma_r$ when applied to the KG $\mathcal{K}^\sigma_r$.

In general, verifying whether a completion function captures a rule requires checking an infinite number of assignments. We can, however, restrict ourselves to a small finite number and rely on the following proposition (proven in the appendix) when comparing, in Section 4.5, the ability of different approaches to capture rules in practice, provided the completion function $f$ realised by a system under consideration is *constant-agnostic*—that is, such that $f(\mathcal{K}, \lambda) = f(\varrho(\mathcal{K}), \varrho(\lambda))$ for every KG $\mathcal{K}$, candidate triple $\lambda$, and renaming $\varrho$ of constants. As far as we could check, all the systems considered in this paper realise constant-agnostic completion functions.

**Proposition 1.** *Let $r$ be a rule of form* (1)*, let $C$ be a set of $|\mathbf{x}|$ constants, and let $\Sigma$ be the set of all assignments of constants from $C$ to $\mathbf{x}$. A constant-agnostic completion function $f$ captures $r$ if and only if $f(\mathcal{K}^\sigma_r, t^\sigma_r)$ is true for each $\sigma \in \Sigma$.*

## 4 Evaluation

We have implemented our approach using Python and PyTorch v1.4.0 in a system called INDIGO. We used R-GCN [17], GraIL [21], and the system by Hamaguchi et al. [5] as baselines. We also tried to include the system of Wang et al. [24], but we found that the system crashes during testing. All experiments were performed on an Intel(R) Xeon(R) machine with 8 cores and a 2.6 GHz CPU equipped with 540 GB of RAM running Fedora 33 (x86_64).

### 4.1 Benchmarks

We exploit a number of benchmarks proposed by Teru et al. [21] and Hamaguchi et al. [5] for inductive KG completion. The benchmarks by Teru et al., 12 in total, are based on transductive benchmarks FB15K-237 [22], NELL-995 [26], and WN18RR [4], and have four versions for each of these transductive benchmarks; we will call them using the pattern GraIL-BM / *XXX*.v$i$ where *XXX* is the base and $i$ the number of the version (e.g., GraIL-BM / NELL-995.v3 is the third version of the benchmark based on NELL-995). Similarly, the benchmarks by Hamaguchi et al., 9 in total, are all based on a transductive benchmark WordNet11 [19], and we call them Hamaguchi-BM / $X$-$i$K, where $X$ is one of h, t, b and $i = 1, 3, 5$ (these parameters specify how the benchmark was constructed, but the exact details are not essential for this paper).

Each of these benchmarks provides the following:

- disjoint sets $\mathcal{T}$ and $\mathcal{V}$ of triples with $\mathsf{Sig}(\mathcal{V}) \subseteq \mathsf{Sig}(\mathcal{T})$ for training and validation;

- an incomplete KG $\mathcal{K}_{\text{test}}$ and a set $\Lambda^+_{\text{test}}$ of test triples with $\mathsf{Sig}(\Lambda^+_{\text{test}}) \subseteq \mathsf{Sig}(\mathcal{K}_{\text{test}})$ that are to hold in the completion of $\mathcal{K}_{\text{test}}$ for testing.

These benchmarks implicitly assume that all provided triples represent true facts. In contrast, all other triples over the relevant signature represent facts that are unknown, but assumed to be *pseudo-negative*—that is, are false with equal probability. This relatively weak assumption is due to the fundamental incompleteness and open-world nature of existing KGs, which make it difficult to discern the truth status of a triple not mentioned in the KG when designing a benchmark. The equal probability ensures that no system has an advantage when evaluated on these benchmarks under the assumption that all pseudo-negative triples are truly negative. Furthermore, to capture the inductive setting, triples in all sets use the same types and relations, but the test sets contain constants that are not mentioned in the training and validation sets.

These existing benchmarks are, however, limited in that they capture the KG evolution scenario in Fig. 1 only partially. On the one hand, in benchmarks GraIL-BM the training and test sets mention completely disjoint sets of constants, thus not capturing the situation where triples mentioning both existing and new constants are added to the KG; on the other hand, in benchmarks Hamaguchi-BM each triple in the testing set always uses both a constant mentioned in the training set and an unseen constant, thus not capturing the situation where a KG is also extended with triples mentioning only unseen constants. To address this limitation, we have designed a new benchmark, called INDIGO-BM, which is based on FB15K-237 and has the same structure and assumptions as the existing benchmarks, but where the use of unseen constants in the test sets is not restricted in any specific way. Additionally, in contrast to existing benchmarks, which contain no type information, our benchmark comes equipped with type triples. Further details of the construction are given in the appendix.

The statistics of all our 22 benchmarks are summarised in Table 1.

## 4.2 Performance Metrics

We evaluate the systems' performance using standard classification metrics (e.g., accuracy and F1-score) and ranking metrics (e.g., Hits@$k$), which are computed based on the systems' outcomes on sets $P$ and $N$ of positive and negative examples. For a benchmark with incomplete KG $\mathcal{K}_{\text{test}}$ and a set $\Lambda_{\text{test}}^+$ of test triples, we take $P$ as the set of all pairs $(\mathcal{K}_{\text{test}}, \lambda)$ with $\lambda \in \Lambda_{\text{test}}^+$ for all metrics; in turn, we obtain $N$ by sampling from the set $N^*$ of all pairs $(\mathcal{K}_{\text{test}}, \lambda)$ with the pseudo-negative triples $\lambda \notin \mathcal{K}_{\text{test}} \cup \Lambda_{\text{test}}^+$ and $\text{Sig}(\lambda) \subseteq \text{Sig}(\mathcal{K}_{\text{test}})$ using different sampling methods for classification and ranking-based metrics as described below. Note that sampling is necessary because $N^*$ is usually very large, and so using all its triples is infeasible. To mitigate the effects of possible fluctuations caused by sampling, we evaluate each system on a given benchmark over 10 runs with independently sampled sets of negative examples, and report the average and variance for each metric.

For classification-based metrics, we construct $N$ by randomly sampling, with equal probability, one element of $N^*$ for each positive example in $P$. This method ensures that systems cannot gain advantage by adopting a particular sampling strategy for negative examples during training [12]. Classification-based metrics for a KG completion system evaluated on $P$ and $N$ are then computed in the standard way based on the numbers $tp, tn, fp, fn$ of true positives, true negatives, false positives, and false negatives. In our experiments, we use *precision* $tp/(tp + fp)$, *recall* $tp/(tp + fn)$, and *accuracy* $(tp + tn)/(tp + tn + fp + fn)$. Besides this, we use *F1-score* and *the area under the precision-recall curve* (*AUC*), which are defined as usual from precision and recall for different thresholds (computed using confidence-based predictions).

For *ranking-based* metrics, we construct $N = N_{\text{c}} \cup N_{\text{r}} \cup N_{\text{d}}$ by randomly "corrupting" positive examples. In particular, for each positive example $(\mathcal{K}_{\text{test}}, (c, r, d))$ in $P$, set $N_{\text{c}}$ contains 50 randomly sampled negative examples in $N^*$ of the form $(\mathcal{K}_{\text{test}}, (c', r, d))$; sets $N_{\text{r}}$ and $N_{\text{d}}$ are constructed analogously by corrupting $r$ and $d$ and taking all and 50 samples, respectively (note that the number of candidates for $N_{\text{r}}$ is bounded by the number of relations in $\text{Sig}(\mathcal{K}_{\text{test}})$). Our ranking-based metrics for a KG completion system evaluated on $P$ and $N$ are then computed as follows. For each $\lambda \in \Lambda_{\text{test}}^+$ and $x \in \{\text{c}, \text{r}, \text{d}\}$, let $Rank_x(\lambda)$ be the position of $(\mathcal{K}_{\text{test}}, \lambda)$ in the decreasing-ordered list of the system's confidence predictions for all the examples in $P \cup N_x$. Then, we use *entity* and *relation hit metrics* e-Hits@$k$ = $(\text{Hits}_{\text{c}}@k + \text{Hits}_{\text{d}}@k)/2$ and r-Hits@$k$ = $\text{Hits}_{\text{r}}@k$, for $k = 1, 3, 10$, where, for each $x$, $\text{Hits}_x@k = m/|\Lambda_{\text{test}}^+|$ for $m$ the number of $\lambda \in \Lambda_{\text{test}}^+$ such that $Rank_x(\lambda) \leq k$. We also use the *entity* and *relation mean reciprocal ranks* e-MRR = $(\text{MRR}_{\text{c}} + \text{MRR}_{\text{d}})/2$ and r-MRR = $\text{MRR}_{\text{r}}$, where, for each $x$, $\text{MRR}_x = h/|\Lambda_{\text{test}}^+|$, for $h$ the sum of $1/Rank_x(\lambda)$ for all $\lambda \in \Lambda_{\text{test}}^+$.

Table 2: Main metric results on the benchmarks in %, where R, G, H, and I stand for R-GCN, GraIL, the system of Hamaguchi et al., and INDIGO, respectively.

| Bench-mark | | Accuracy | | | | AUC | | | | e-Hits@3 | | | | r-Hits@3 | | | |
|---|---|---|---|---|---|---|---|---|---|---|---|---|---|---|---|---|---|
| | | R | G | H | I | R | G | H | I | R | G | H | I | R | G | H | I |
| GraIL-BM FB15K-237 | v1 | 51.0 | 69.0 | - | **84.3** | 51.0 | 78.6 | - | **93.4** | 16.1 | 43.4 | - | **45.1** | 2.4 | 1.0 | - | **53.1** |
| | v2 | 51.3 | 80.0 | - | **89.3** | 50.5 | 90.0 | - | **96.3** | 18.3 | **68.5** | - | 36.2 | 3.4 | 0.4 | - | **67.6** |
| | v3 | 54.9 | 81.0 | - | **89.0** | 50.5 | 93.1 | - | **96.6** | 14.1 | **71.2** | - | 33.9 | 3.5 | 6.6 | - | **66.5** |
| | v4 | 52.1 | 79.3 | - | **87.8** | 52.6 | 89.5 | - | **95.8** | 16.1 | **61.8** | - | 37.1 | 3.3 | 3.0 | - | **66.3** |
| NELL-995 | v1 | 63.7 | **97.3** | - | 85.6 | 74.5 | **98.8** | - | 94.5 | 7.5 | **51.0** | - | 39.5 | 26.0 | 0.0 | - | **80.0** |
| | v2 | 52.0 | 68.5 | - | **84.1** | 50.4 | 89.7 | - | **92.5** | 12.2 | **76.5** | - | 44.2 | 0.8 | 7.4 | - | **56.9** |
| | v3 | 52.3 | 74.3 | - | **89.7** | 52.0 | **95.4** | - | 95.1 | 15.5 | **84.4** | - | 45.0 | 1.4 | 2.5 | - | **64.4** |
| | v4 | 53.6 | 49.7 | - | **85.2** | 51.0 | 65.8 | - | **92.9** | 9.3 | **56.0** | - | 52.3 | 3.0 | 0.5 | - | **45.7** |
| WN18RR | v1 | 50.2 | 88.7 | - | **85.7** | 49.0 | **92.3** | - | 91.2 | 20.1 | **82.7** | - | 12.5 | 2.1 | 0.6 | - | **98.4** |
| | v2 | 52.7 | 81.2 | - | **85.8** | 49.8 | **92.7** | - | 92.5 | 18.1 | **81.5** | - | 18.8 | 11.0 | 10.7 | - | **97.3** |
| | v3 | 52.2 | 75.7 | - | **84.3** | 53.1 | 82.8 | - | **92.4** | 16.9 | **55.5** | - | 33.1 | 24.5 | 17.5 | - | **91.9** |
| | v4 | 48.4 | **86.4** | - | 85.4 | 50.2 | 94.4 | - | **94.7** | 8.8 | **76.3** | - | 13.5 | 8.1 | 22.6 | - | **96.1** |
| Hamaguchi-BM | h-1K | 43.0 | 49.3 | **83.6** | 75.3 | 43.0 | 51.5 | 77.7 | **87.4** | 31.6 | 13.8 | **55.6** | 32.9 | 31.4 | 29.6 | 47.1 | **80.0** |
| | h-3K | 44.1 | 50.2 | 79.0 | **80.8** | 43.0 | 55.6 | 72.4 | **91.2** | 29.7 | 17.7 | **48.1** | 35.3 | 27.8 | 29.1 | 41.6 | **83.8** |
| | h-5K | 46.5 | 50.0 | 79.7 | **83.3** | 45.4 | 56.8 | 73.1 | **93.4** | 23.9 | 21.0 | **49.7** | 33.4 | 31.3 | 26.1 | 38.5 | **86.2** |
| | t-1K | 49.8 | 53.5 | 77.1 | **81.8** | 48.7 | 58.8 | 71.4 | **90.3** | 17.1 | 23.9 | **42.9** | 31.0 | 27.6 | 31.7 | 43.3 | **85.4** |
| | t-3K | 42.5 | 53.1 | 75.3 | **85.8** | 41.4 | 61.7 | 69.2 | **94.3** | 24.9 | 19.5 | 31.6 | **33.9** | 30.3 | 27.9 | 35.3 | **88.0** |
| | t-5K | 40.8 | 53.4 | 74.0 | **87.0** | 39.2 | 62.0 | 67.7 | **95.3** | 21.3 | 26.7 | 31.6 | **36.1** | 25.4 | 25.5 | 37.4 | **88.3** |
| | b-1K | 22.4 | 54.0 | 85.0 | **87.0** | 33.4 | 59.2 | 79.2 | **95.4** | **62.5** | 26.5 | 46.7 | 43.2 | 19.8 | 32.1 | 38.5 | **93.8** |
| | b-3K | 27.3 | 53.8 | 79.3 | **86.8** | 35.4 | 61.1 | 72.1 | **95.7** | 36.0 | 22.3 | 37.1 | **37.5** | 20.2 | 28.6 | 41.5 | **90.1** |
| | b-5K | 28.3 | 52.6 | 75.8 | **88.8** | 35.1 | 58.8 | 68.4 | **96.8** | 24.3 | 26.3 | 30.6 | **39.1** | 22.1 | 27.2 | 34.0 | **92.0** |
| INDIGO-BM | | 73.9 | 86.2 | - | **94.4** | 89.5 | 94.2 | - | **99.0** | 36.1 | **73.6** | - | 53.5 | 32.4 | 4.9 | - | **77.0** |

## 4.3 Training

Our INDIGO system is trained as a *denoising autoencoder* [23]. The training set $\mathcal{T}$ of a benchmark is first randomly split, with ratio 9:1, into an incomplete KG $\mathcal{K}_{\text{train}}$ and a set $\Lambda^+_{\text{train}}$ of triples assumed to hold in the completion of $\mathcal{K}_{\text{train}}$. Then, the model is trained on positive examples $(\mathcal{K}_{\text{train}}, \lambda)$ for each $\lambda \in \Lambda^+_{\text{train}}$ and negative examples $(\mathcal{K}_{\text{train}}, \lambda)$ for $\lambda$ sampled from the set $\mathcal{N}$ of triples not in $\mathcal{T}$ using the strategy described next. For each triple $(c, r, d) \in \Lambda^+_{\text{train}}$ with $r \neq$ type we sampled (with equal probability) the following triples from $\mathcal{N}$: three triples of the form $(c, r', d)$ where $r'$ is *disjoint* from $r$ role in $\mathcal{T}$—that is, such that $\mathcal{T}$ has no triples $(c', r, d')$ and $(c', r', d')$; three triples of the form $(c, r, d')$, if $r$ is *functional* in $\mathcal{T}$—that is, $\mathcal{T}$ has no triples $(c', r, d_1)$, $(c', r, d_2)$ with $d_1 \neq d_2$; and three triples of the form $(c', r, d)$, if $r$ is *inverse-functional* in $\mathcal{T}$—that is, $\mathcal{T}$ has no triples $(c_1, r, d')$, $(c_2, r, d')$ with $c_1 \neq c_2$. For each triple $(c, \text{type}, t) \in \Lambda^+_{\text{train}}$ we similarly sampled from $\mathcal{N}$ three triples of the form $(c, \text{type}, t')$ where $t'$ is *disjoint* from $t$ type in $\mathcal{T}$, which is defined analogously to disjoint roles.

As a result, we generate up to 9 negative training examples for each positive example.

We employed the standard cross-entropy loss function and trained for 3,000 epochs using Adam optimisation with L2 penalty 5e-8. We set as hyper-parameters the number of layers in the GCN (2, 3, or 4), the dimension of vectors in the hidden layers (32, 64, or 128) and the learning rate (0.01 or 0.001) and cross-validated them on each of the benchmarks using the validation sets to obtain a most favourable (for all benchmarks) setting of 2 layers, vector dimension of 64, and learning rate 0.001.

Our baseline systems are trained similarly as autoencoders. We trained them using their (publicly available) code and the settings reported in the papers without modifications, including their negative sampling strategies for training, the values of hyper-parameters, and the number of training epochs.

Table 3: Results on capturing high-confidence rules, where each entry indicates both the number of the high-confidence rules of the benchmark represented by the pattern that are captured by the model and the percentage of the total number of the represented rules.

| Rule pattern | GraIL-BM / NELL-995.v3 | | | INDIGO-BM | | |
|---|---|---|---|---|---|---|
| | R-GCN | GraIL | INDIGO | R-GCN | GraIL | INDIGO |
| $(x, \_\mathtt{r}, y) \to (x, \_\mathtt{s}, y)$ | - | - | - | 2  (9%) | **3 (13%)** | **3 (13%)** |
| $(x, \mathtt{type}, \_\mathtt{t}) \to (x, \mathtt{type}, \_\mathtt{u})$ | - | - | - | 89 (20%) | 0 | **207 (46%)** |
| $(x, \_\mathtt{r}, y) \to (y, \_\mathtt{r}, x)$ | 0 | 0 | **4 (80%)** | 8 (33%) | 0 | **14 (58%)** |
| $(x, \_\mathtt{r}, y), (y, \_\mathtt{s}, z) \to (x, \_\mathtt{t}, z)$ | 0 | **2 (12%)** | **2 (12%)** | 95 (14%) | 130 (19%) | **150 (22%)** |
| $(x, \_\mathtt{r}, y), (x, \_\mathtt{s}, y) \to (x, \_\mathtt{t}, y)$ | 0 | 0 | **42 (49%)** | 19  (7%) | 19  (7%) | **116 (41%)** |
| $(x, \mathtt{type}, \_\mathtt{t}), (x, \mathtt{type}, \_\mathtt{u}) \to (x, \mathtt{type}, \_\mathtt{v})$ | - | - | - | 5281 (20%) | timeout | **12288 (47%)** |

## 4.4 Evaluation Results

The results for several indicative metrics (accuracy, AUC, e-Hits@3, and r-Hits@3) of the metric-base evaluation of the systems on the benchmarks are summarised in Table 2 (the system by Hamaguchi et al. is not applicable to GraIL-BM and INDIGO-BM due to its limited inductive capabilities); results for other metrics are given in the appendix. As we can see, INDIGO consistently outperforms, often by a significant margin, the baselines on almost all metrics. A notable exception is ranking-based entity metrics e-Hits@$k$ and e-MRR, where INDIGO is often worse than the baselines. This can be explained by the fact that the baselines use entity corruption in negative sampling for training, which can be seen as a bias towards these metrics, while our sampling strategy in training is not prejudiced to any metric. In addition to achieving better results, our system is also significantly faster to train. For instance, training on Hamaguchi-BM/b-5k dataset took more than 0.5 hours for R-GCN, more than 4.5 hours for GraIL, 0.9 hours for the system of Hamaguchi et al., and 0.37 hours for INDIGO in the reported configurations. Detailed time statistics for training and testing is given in the appendix.

## 4.5 Capturing Logical Rules

We also studied the systems' ability to learn rules by means of capturing, as discussed in Section 3.5. In particular, we considered several commonly used rule patterns, which are specified in the first column of Table 3. Each pattern represents all rules obtained by appropriately substituting its templates $\_\mathtt{a}$ by types and relations. We took GRAIL-BM/NELL-995.v3 and INDIGO-BM benchmarks for experiments, and considered INDIGO, GraIL and R-GCN trained for these benchmarks. For each benchmark with an incomplete KG $\mathcal{K}_{\mathsf{test}}$ and a set $\Lambda_{\mathsf{test}}^+$ of test triples, and for each rule pattern we proceeded as follows. We first generated all rules represented by the pattern over the types and relations in $\mathsf{Sig}(\mathcal{K}_{\mathsf{test}})$. We then identified the rules with confidence at least 0.7, where the *confidence* of a rule $r$ in $\mathcal{K}_{\mathsf{test}}$ and $\Lambda_{\mathsf{test}}^+$ is $n/m$ for $m$ the number of assignments $\sigma$ such that $K_r^\sigma \subseteq \mathcal{K}_{\mathsf{test}}$ and $n$ the number of such $\sigma$ that also satisfy $t_r^\sigma \in \mathcal{K}_{\mathsf{test}} \cup \Lambda_{\mathsf{test}}^+$. Finally, for each model we computed, using Proposition 1, the proportion of captured high-confidence rules.

Our results are summarised in Table 3. As we can see, INDIGO was consistently able to capture the highest number of high-confidence rules for each of the patterns. This suggests that our system is able to inductively generalise its predictions more effectively than the baselines.

## 5  Conclusion and Future Work

In this paper we have proposed a novel GNN-based approach to inductive KG completion which significantly outperforms state-of-the-art approaches. The key novelty of our approach is that it encodes KGs using a one-to-one correspondence between triples in the KG and elements of nodes' feature vectors in the graphs processed by the GNN. A current limitation of our approach is that all predicted triples involve only the relations and types mentioned in the original KG; there has been recent work on methods that are able to complete the KG with triples over unseen relations and types [13], and it would be interesting to see if our techniques can be extended to this setting. For future work, we are also planning to explore the potential applications of our approach in real-world settings (such as KG-enhanced recommendation systems) in collaboration with our industrial partner.

## Acknowledgments and Disclosure of Funding

This work was supported by the EPSRC projects OASIS (EP/S032347/1), UK FIRES (EP/S019111/1), the SIRIUS Centre for Scalable Data Access, and Samsung Research UK.

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
