# OpenReview forum: "INDIGO: GNN-Based Inductive Knowledge Graph Completion Using Pair-Wise Encoding"
_NeurIPS.cc/2021/Conference — NeurIPS 2021 Poster_

### Official Review · Reviewer_WdL8 · 2021-07-14

**Rating:** 5
**Confidence:** 4

**Summary:**

This paper proposes a novel approach called INDIGO for KG completion in the inductive setting. INDIGO is based on GNNs, and it adopts a distinct way to encode the KG into the input graph of a GNN. In the well-designed input graph, each node represents a pair of constants (i.e., entities), and two nodes are connected only if their pairs share a constant. Meanwhile, each node has an initial feature vector where each element captures a fact involving the pair constants in the KG. After the training of GNNs, the predicted triples can be read out directly based on the output representations from the last layer. The proposed method overcomes the drawbacks of the existing scoring-based approaches that ignore the common part of neighborhoods of entities, and can efficiently make predictions for many triples at one time. Experiments on several inductive KG completion benchmarks demonstrate the effectiveness of the proposed method.

**Limitations And Societal Impact:**

It seems that INDIGO needs to know the candidate set in advance, as it needs to encode the KG and the candidate set together into the input graph of a GNN. Besides, if some changes happen in the candidate set, INDIGO may need retraining. I think these two issues are the limitations of INDIGO.

**Main Review:**

**Clarity**:

Generally, I think this paper is clearly written, especially in Section 3 to describe the proposed architecture. However, there is still room for improvement in the organization of the paper. For example, the analyses of the experimental results are somewhat limited, and I think the description of benchmarks and performance metrics seems a bit lengthy.

**Originality**:

From my perspective, I think the proposed method on how to encode the KG into the input graph of a GNN sounds interesting and inspiring. However, it still lacks some novelty. In the architecture, it just uses a GCN to update the feature vectors and simply adopts a threshold to decode the output. I think more careful designs on these modules will make this work more solid. For example, [1] argues that a sigmoid layer to turn logits into probabilities will provide poor calibration, so the output of the proposed method may also need a well-designed calibration before adopting 0.5 as the threshold.

**Quality**:

In general, I think this paper is technically sound. But, I still have a few questions and suggestions on experimental results. First, I notice that the authors argue existing benchmarks have some issues. They claim that some benchmarks cannot capture the situation where the triples mentioning both existing and new constants are added to the KG, and the others cannot capture the situation where a KG is also extended with the triples mentioning only unseen constants. By contrast, their proposed benchmark can capture both situations. However, the detailed statistics on the two situations are not listed, and it will be better to provide evaluation results on these two situations respectively. I think this will help the readers know the difficulty of the two situations, as well as the effectiveness of their proposed method and baselines on the two situations. Second, I notice that the proposed benchmark contains type information, and INDIGO can use such information while other baselines seem not to use such type information. So, is it a fair comparison?

**Other questions**:

The authors say that their proposed method can be trained and applied efficiently, and the results also show that INDIGO spends much less time on training. But I think the scale of the input graph in the proposed method is much larger than that in baselines, as each pair of constants may be a node. So the number of nodes in the input graph may reach O(N^2) (N is the number of entities). Due to this issue, I want to know why INDIGO is so efficient compared with other baselines.

[1] Pedro Tabacof, Luca Costabello: Probability Calibration for Knowledge Graph Embedding Models. ICLR 2020

**Time Spent Reviewing:**

2

---

> ### Author Response · Authors · 2021-08-09
> **Authors' response to Review of Paper8745 by Reviewer WdL8**
>
> We thank the reviewer for their thoughtful feedback.
>
>
>
> Comment:
> However, it still lacks some novelty. In the architecture, it just uses a GCN to update the feature vectors and simply adopts a threshold to decode the output. I think more careful designs on these modules will make this work more solid. For example, [1] argues that a sigmoid layer to turn logits into probabilities will provide poor calibration, so the output of the proposed method may also need a well-designed calibration before adopting 0.5 as the threshold.
>
> -- Answer:
> We highlight here that our main contribution in this paper is essentially a specially-designed encoding/decoding of the KG. Thus, we adopted sigmoids and threshold 0.5 for simplicity, and achieved good performance already here. It is easy to use other non-linearities and thresholds in our approach; in fact, we tried the average of the output score as a threshold instead of a fixed 0.5, but
> obtained slightly worse performance.
>
>
>
>
> Comment:
> However, the detailed statistics on the two situations are not listed, and it will be better to provide evaluation results on these two situations respectively. I think this will help the readers know the difficulty of the two situations, as well as the effectiveness of their proposed method and baselines on the two situations.
>
> -- Answer:
> Please note that, as described in Section 4.1 (Lines 272-282) the three groups of benchmarks we use already test the systems precisely in these situations: in the benchmark by Teru et al. [20] the training and test sets mention completely disjoint sets of constants; in the benchmark by Hamaguchi et al. [5] each triple in the testing set always uses both a constant mentioned in the training set and an unseen constant; our INDIGO-BM benchmark is a mixture of these situations.
>
>
>
>
> Comment:
> I notice that the proposed benchmark contains type information, and INDIGO can use such information while other baselines seem not to use such type information. So, is it a fair comparison?
>
> -- Answer:
> As mentioned  in Section 4.1 (Line 281) only our INDIGO-BM benchmark contains type information, while the benchmarks proposed by Teru et al. [20] and Hamaguchi et al. [5], where we also obtain
> very favourable performance, do not mention type triples.
> Please note also that the baseline systems do take into account type information, by treating type triples as ordinary triples (and hence by treating "type" as an ordinary relation).
> We consider our special treatment of type information as a feature and (minor) improvement of the state of the art.
>
>
>
>
> Comment:
> The authors say that their proposed method can be trained and applied efficiently, and the results also show that INDIGO spends much less time on training. But I think the scale of the input graph in the proposed method is much larger than that in baselines, as each pair of constants may be a node. So the number of nodes in the input graph may reach O(N^2) (N is the number of entities). Due to this issue, I want to know why INDIGO is so efficient compared with other baselines.
>
> -- Answer:
> According to Section 3.2 (Lines 142-144) the number of nodes in the new graph
> is bounded by the number $|K|$ of triples in the original graph plus the number $|\Lambda|$ of candidate triples, so it is linear in the size of the input.
> This indeed may be quadratic in the number of entities (i.e., constants and types) in theory, but this is not a limitation in practice, due to the linear bound
>  (see also our response to Reviewer 2, in particular to the related comment
>  starting with "Definition 2...").
>
>
>
>
> Comment:
> It seems that INDIGO needs to know the candidate set in advance, as it needs to encode the KG and the candidate set together into the input graph of a GNN. Besides, if some changes happen in the candidate set, INDIGO may need retraining.
>
> -- Answer:
> Please note that, as any other inductive KG completion system, INDIGO needs to know only the training examples when training (where training examples include their labelled candidate triples, as expected). Once a model is trained, it can be applied--that is, tested--to
> any graph and any appropriate candidate set without re-training (see also the definition of Inductive completion problem in Section 2).

---

> > ### Comment · Reviewer_WdL8 · 2021-09-12
> > **Thanks for the detailed response**
> >
> > Thank the authors for their detailed response.
> >
> > I still have a concern regarding the novelty. But it may not be a serious weakness since the proposed method seems to work well.

---

### Official Review · Reviewer_CYvG · 2021-07-18

**Rating:** 6
**Confidence:** 4

**Summary:**

Knowlegde graph completion (KGC) has traditionally been attempted in transductive mode: a fixed (incomplete) graph is provided, the algorithm analyzes it, and proposes edges between existing nodes.  The inductive version, where new nodes never seen during training are then presented, and (subject, relation, object) queries have to be scored with unseen subject and/or object, is less studied, and is the subject of this paper.  The paper proposes a new encoding/transformation of the training+test graph, new in the KGC task but not particularly striking as far as graph representation goes; it is a variation of a product graph.  Then it proposes to run a fairly standard GCN on top of the transformed graph.  The output from the GCN is used to predict new edges in the original KG.  Experiments show frequent, sometimes dramatic gains over some recent systems.

The core idea (Defn 2) is to build a new graph from the KG, where each node is a pair of (perhaps identical) entities.  Roughly speaking, each node $(e,e)$ in the new graph, where $e$ is an entity in the original KG is associated with a bit vector over the entity types of the KG.  If $e$ is an instance of type $t$, then the $t$th bit is set.  Also, each node pair $(s,o)$ in the new graph, where $(s,r,o)$ is in the original KG, is associated with a bit vector over the relation types of the KG, with the $r$th bit set.   Two nodes in the new graph are connected by an edge if any of their component entities in the original KG is common.  On this new graph they run a GCN.  The GCN representation of a node is used to infer a softmax over types or relations as appropriate.  This makes the KGC scheme inductive.

**Main Review:**

Novelty and importance of problem spec: Yes, this is an important problem where insufficient work exists.

Originality of technique: This general form of graph construction using a Cartesian space of nodes is not new; it has been applied in link prediction, but I have not seen it applied to KGC.

Effectiveness and impact: The numbers presented seem impressive, but I am not familiar with the latest numbers on most of these specific data sets.  Provided one question I raise about the size of the derived graph is satisfactorily answered, the technique seems effective.


Detailed comments:

L11 "external heuristics and ad-hoc scoring functions" --- unclear.

L24 entity + "and relation"

L26 Throughout, "constants" is used in a sense not that common in KG literature. If you wish to use "constants" throughout, define it precisely at this point.

In Figure 1 caption, explain that G, X, P, S are shorthand. Explain what $Col_{1,2}$ mean.  Explanatory text is too far; do this in the caption itself.

One feature of your approach that you can highlight better is the collective nature of inference.  In standard KGC, inference (scoring) for (s,r,o) triples is done one by one.  Your approach allows collective inference of a whole test batch.

L41 is -> are

L52 "randomly initialized" --- why?  Only if there is no usable local info at nodes, which is never the case with KGs.  Nodes have rich aliases, descriptions, attribute names and values.  These have been used to get very good baseline accuracies in BERT-INT, for example.

[BERT-INT] https://www.ijcai.org/Proceedings/2020/0439.pdf

L57 "does not take into account what is the common part of these neighbourhoods"

remove "what is"

Apart from the fixes you cite, also see
https://proceedings.neurips.cc/paper/7763-link-prediction-based-on-graph-neural-networks.pdf and https://arxiv.org/abs/2005.00687 for more pointers.

L67 "one to one correspondence" --- completely unclear until one reads and understands your technique.

Definition 2 must be worked better into your intro --- it's a salient contribution.

L80 unclear what you mean by "pairwise disjoint" here.  Do you mean "mutually exclusive"?  That is never the case with real KGs.

L83 (c,r,d) is strange notation; (s,r,o) = (subject, relation, object) or (h,r,t) = (head, relation, tail) is commonly used.

L84 "the subset" -- which subset?

L106 "Definition 1" -- this is too much of a forward reference.  Besides, it does not define much!

L109 The limitation of GNCs are well documented. Why not use a more advanced member of the GNN family, evolved in recent years?

Definition 2 is the core of the proposal.  Here (or much earlier), you need to clarify one point that may occur to the reader.  What is the number of nodes and edges in the new graph you construct, as functions of the number of entities, types, and relations in the original KG?  E.g., in Figure 1 itself, we see that R-GCN uses a graph with 4 nodes and 5 edges, whereas your representation uses 9 nodes and at least 20 edges.

I scanned Table 1 to look for a direct comparison between the number of nodes in the original KG and the derived graph, as well as the number of edges.  But I could not get a direct idea of the typical blow-up you would suffer.  Particularly in view of your spectacular claims around L348 (and appendix tables 7, 8), you need to explain how you control the potentially explosive growth in the number of nodes and edges.

L186 Section 3.3 Before what the GNN model is, explain WHY you wish to run a GNN over the newly constructed graph, i.e., a brief verbal preview of section 3.4.

L213 "elements" -> "trainable weights"

Table 1/ L229 --- you should really use more descriptive and self-contained captions --- if they run into 5 lines, that's perfectly fine.  Remind what $\mathcal{T}, \mathcal{V}, \mathcal{K}_\text{test}, \Lambda_\text{test}$ mean.  Don't just present the numbers, interpret them and comment on them.  Coded names like v1 \dots v4 and h-1K t-5K etc. must be locally explained.

L307 What is the sensitivity of MRR to the number of negatives you sample per positive?


**Time Spent Reviewing:**

4 hours

---

> ### Author Response · Authors · 2021-08-09
> **Authors' response to Review of Paper8745 by Reviewer CYvG**
>
> We thank the reviewer for the efforts in reviewing our paper and for their helpful comments.
> We will address their suggestions in the next version of the paper.
>
>
> Comment:
> This general form of graph construction using a Cartesian space of nodes is not new; it has been applied in link prediction, but I have not seen it applied to KGC.
>
> -- Answer:
> We are not aware of any work that adopts this construction for link prediction. We would be very
>  grateful if the reviewer could provide a reference so we can compare it with our work.
>
>
>
>
>
> Comment:
> Throughout, "constants" is used in a sense not that common in KG literature. If you wish to use "constants" throughout, define it precisely at this point.
>
> -- Answer:
> Please note that the notion of a constant is formally defined in Section 2 (Line 81): it is an element of the set of constants in the signature, and does not have any further essential properties or structure.
>
>
>
>
> Question:
> L52 "randomly initialized" --- why? Only if there is no usable local info at nodes, which is never the case with KGs. Nodes have rich aliases, descriptions, attribute names and values. These have been used to get very good baseline accuracies in BERT-INT, for example.
>
> -- Answer:
> Please note that in this sentence we talk about a specific approach proposed in the literature,
> R-GCN [17], which uses random initialisation. (The R-GCN paper
> [17] does not discuss initialisation, but the fact that it is random can be easily checked
> by inspection of their publicly available source code.)
>
>
>
>
> Question:
> unclear what you mean by "pairwise disjoint" here. Do you mean "mutually exclusive"? That is never the case with real KGs.
>
> -- Answer:
> As standard in set theory, two sets are disjoint if they have no elements in common, and several sets are pairwise disjoint if every two different of them are disjoint.
> Practical KG formats, such as RDF, indeed allow for a same element (e.g., IRI) to be used as both a relation and an entity. However, this can be easily avoided by
> renaming apart the occurrences of such elements in different roles,
> and thus such normalised form is often assumed in the literature on RDF and knowledge graphs.
>  In fact, the KGs in all benchmarks we use already have these sets disjoint. In any case, our results do not depend on this assumption or renaming, and 'pairwise disjoint' may be safely removed from the paper without any further changes.
>
>
>
>
> Question:
> The limitation of GCNs are well documented. Why not use a more advanced member of the GNN family, evolved in recent years?
>
> -- Answer:
> We highlight here that our main contribution in this paper is essentially a specially-designed encoding/decoding of the KG. Thus, in the GNN part, we adopted standard GCNs for simplicity, and achieved good performance already here.
> GCNs in our architecture could be easily replaced with other GNN models.
>
>
>
> Comment:
> Definition 2 is the core of the proposal. Here (or much earlier), you need to clarify one point that may occur to the reader. What is the number of nodes and edges in the new graph you construct, as functions of the number of entities, types, and relations in the original KG?… you need to explain how you control the potentially explosive growth in the number of nodes and edges.
>
> -- Answer:
> Please note that according to the first sentence of Section 3.2 (Lines 142-144), the number of nodes in the new graph is bounded by the number $|K|$ of triples in the original graph plus the number $|\Lambda|$ of candidate triples. In theory, $|\Lambda|$ can be quadratic in $|K|$, but in all benchmarks
> we know of $|\Lambda|$ is much smaller than $|K|$ (see Table 1 and Section 4.2). As for number of edges in the new graph, it can be theoretically bounded by $(|K|+|\Lambda|) * A$, where $A$ is the maximal degree of a constant
>  in the original KG--that is, the number of triples the constant appears. In theory, $(|K|+|\Lambda|) * A$ can be as much as ($|K|+|\Lambda|)^{3/2}$; however, this bound is approached only if every two constants are connected by a triple in $K$ or $\Lambda$.
>  This is never the case in practice since KGs are typically very sparse; as a result,
>   $A$ is small, and we can assume that the number of edges in the new graph is not much larger than $|K|+|\Lambda|$ and hence linear
>   in the size of the input.
> We thank the reviewer for the suggestion and will be glad to add these clarifications to the next version.
>
>
>
>
>
>
> Comment:
> What is the sensitivity of MRR to the number of negatives you sample per positive?
>
> Answer:
> Please note that we took 50 as the number of negatives per positive in our evaluation metrics following the literature [20]. We experimented with other ratios, and, not surprisingly, saw that MRR decreases as the number of negatives grows.
> For example, on GraIL-BM_WN18RR_v1, e-MRR of INDIGO is 26.8, 22.5, 16.4, 13.0 for ratios 20, 30, 40, and 50, respectively. Other systems show very similar trends.

---

### Official Review · Reviewer_e3QF · 2021-07-19

**Rating:** 6
**Confidence:** 4

**Summary:**

Summary: In this study the authors propose INDIGO, a GCN based method for inductive knowledge base completion, i.e. predicting relations between known or unknown constants of a KG. The paper provides a great survey over current related approaches and also proposes a new dataset for this setting. The proposed method basically does label (feature) propagation with a GCN, in which the labels (features) are indicators of the relations and the types of the nodes. To induce the presence of a new triple, the hidden state after label (feature) propagation is used for binary predictions, i.e. a pair of constants (c, d) will predicted to be connected by relation r if their labels (features) contain r and and many other pairs of constants that also are in relation r are highly similar regarding the other labels (features). The experiments on two previously proposed datasets and its own dataset show that the proposed method improves in most classification and ranking metrics over previous approaches, except for the entity ranking metric.

**Limitations And Societal Impact:**

Some limitations and impact have been discussed.

**Main Review:**

Originality: None of its elements is new or modified in a novel way, but as far as I am aware it was not applied in this way on knowledge graphs. The work positions itself well in the context of related work.

Quality: The work seems sound. The claims are supported by experimental results, but no theoretical guarantees on the limits of this approach are given and the weaknesses of the approach are not treated completely transparently. The discussion why this is bad for ranking but works apparently better for classification where the negatives to which the true triple is compared against are semantically less close to the true triple is lacking. The approach looks to me like a high recall low precision method. This aspect is somewhat interesting, if it would have been investigated more closely (theoretically or empirically), i.e. what is the limiting factor, how could it be improved? Was it a training parameter like the number of pseudo-negatives or is this as good as it gets? As the paper is about the inductive scenario it would have been necessary to be explicit in the evaluation about which parts of the model need to be updated when new constants are added and what complexity this incurs. Also, isn’t another limiting factor the number of the constants, i.e. the Adjacency matrix? This should be discussed as well also in contrast to the other approaches. As the evaluation is based on metrics computed against random samples it is unclear how stable these results are.

Clarity: The authors use a different language than typical Neurips papers and hence it takes more effort than necessary to follow the paper. The core idea of the paper can be summed up pretty briefly and the paper takes a lot of time to explain the idea. However, I did like the introduction and how the paper contrasts itself with related work and the paper is also quite self-sustained due to its verboseness which is also a good attribute.

Significance: There are some interesting aspects of this work that could inspire follow-ups and it does establish some good results on benchmark datasets and provides another one which is well motivated.


**Time Spent Reviewing:**

2

---

> ### Author Response · Authors · 2021-08-09
> **Authors' response to Review of Paper8745 by Reviewer e3QF**
>
> We thank the reviewer for the positive reviews as well as the insightful questions.
>
>
> Comment:
> Originality: None of its elements is new or modified in a novel way
>
> -- Answer:
> The main novelty of this paper is the encoding/decoding of the KG. This is explained,
> for example, in the Introduction (Lines 65-69) and Conclusion (Lines 371-373).
>
>
>
> Comment:
> No theoretical guarantees on the limits of this approach are given and the weaknesses of the approach are not treated completely transparently.
>
> -- Answer:
> The weaknesses/limitations of our approach are discussed in the Conclusion (Lines 373-376).
> It was unclear to us the particular type of theoretical guarantees the reviewer is referring to; we would  be
> happy to comment should the reviewer provide further clarification.
>
>
> Comment:
> The discussion why this is bad for ranking but works apparently better for classification where the negatives to which the true triple is compared against are semantically less close to the true triple is lacking.
>
> -- Answer:
> Please note that, according to our experiments, our system's performance is weaker
> only for the ranking of *entities* (e-Hits@k and e-MRR metrics), but it outperforms all other systems
> in the ranking of *relations* (r-Hits@k and r-MRR).
> Moreover, as discussed in Section 4.4 (Lines 345-348),
> the unfavourable results concerning the e-Hits and e-MRR metrics can be explained
> by the fact that the baselines use entity
> corruption in negative sampling for training, which can be seen as a bias towards these metrics, while our sampling strategy in training does not favour
> any particular metric.
>
>
>
> Question:
> The approach looks to me like a high recall low precision method. What is the limiting factor, how could it be improved?
>
> -- Answer:
> Please note that, as shown in Table 4 in the appendix, our precision scores tend to be
> higher than our recall scores.
>
>
>
> Question:
> Was it a training parameter like the number of pseudo-negatives or is this as good as it gets?
>
> -- Answer:
> Please, observe (Section 4.2) that the number of sampled pseudo-negatives is not a training parameter but a parameter
> used to compute performance metrics on a benchmark (i.e., a system should not and could not know this parameter when training).
> Following Teru et al., we sampled 50 pseudo-negatives per positive example (for ranking-based metrics); we also experimented with
> other values, and the results were very similar.
>
>
>
>
> Comment:
> As the paper is about the inductive scenario it would have been necessary to be explicit in the evaluation about which parts of the model need to be updated when new constants are added and what complexity this incurs.
>
> -- Answer:
> Please note that the model does not need to be updated in any way when new constants are added.
> As described in Section 2, Lines 88-91,
> the aim in inductive KG completion is to learn a Boolean completion function f (·,·) applicable to any KG
> and triple over a given set of relations and types (but with unrestricted use of constants).
> Thus, an inductive model is trained on KGs over one set of constants and applicable to a KG over other constants without any updates to the model (i.e., without re-training).
>
>
>
> Question:
> Also, isn’t another limiting factor the number of the constants, i.e. the Adjacency matrix?
>
> -- Answer:
> The main limiting factor is training time, which generally increases as the
> the number of examples and the sizes of KGs (i.e., the number of triples) involved increase;
> these are obviously dependent on the number of constants mentioned during training.
> Please, see Tables 7 and 8 in the Appendix, which demonstrate that INDIGO shows
>  its superior results over other systems in generally shorter training time.
>
>
>
>
> Comment:
> As the evaluation is based on metrics computed against random samples it is unclear how stable these results are.
>
> -- Answer:
> As stated in Section 4.2 (Lines 294-296), to mitigate the effects of possible fluctuations caused by sampling, we evaluated
>  each system on a given benchmark over 10 runs independently and reported the average and variance for each metric. Please see the variance in Table 6 in the Appendix. As we can see, the results remain stable when the negative examples change.

---

> > ### Comment · Reviewer_e3QF · 2021-09-02
> > **Thanks for your answers**
> >
> > Dear authors,
> >
> > thank you for your comments.
> >
> > Just two comments to your rebuttal:
> >
> > Your answer is that the negative sampling strategy was at fault for the bad performance in the e-HITS@3 metric. If so, then you could train the model with another negative sampling strategy to support that claim. Even if the system would by design not perform well for all metrics this would not matter but it has to be clearly discussed. This is also related to my comment if the system maybe has theoretical limits by design, this is left unclear by your current paper, but the question jumps at the reader looking at the results.

---

> > > ### Author Response · Authors · 2021-09-02
> > > **Response to Comment of Reviewer e3QF**
> > >
> > > We thank the reviewer for their comment.
> > >
> > > Upon examination of the baselines’ source code, it is clear that they are strongly biased towards the e-HITS@k metrics since their negative sampling strategy during training closely mimics the way the metric is computed. We could, in principle, introduce a similar bias to our system, and we are confident that this would improve our scores for those particular metrics to values competitive with the baselines. Our goal in this paper, however, was to develop a system that performs well when taking into account all relevant metrics rather than a particular one, and which does not exploit unnecessary biases. We believe that our results show that this objective has been achieved.

---

### Decision · Program_Chairs · 2021-09-27

**Decision:**

Accept (Poster)

**Comment:**

This paper describes a new knowledge graph completion alg. which can work on novel entities.  The evaluation is thorough and extensive.  There were several points brought up by the reviewers, and most handled well in the response.

One reviewer thought there should be an additional analysis with a different negative sampling method. The authors respond "Our goal in this paper, however, was to develop a system that performs well when taking into account all relevant metrics rather than a particular one, and which does not exploit unnecessary biases. We believe that our results show that this objective has been achieved." I agree with the authors on this point -- an additional model is not needed